# LC-ESI-MS/MS Identification of Biologically Active Phenolics in Different Extracts of *Alchemilla acutiloba* Opiz

**DOI:** 10.3390/molecules27030621

**Published:** 2022-01-18

**Authors:** Katarzyna Dos Santos Szewczyk, Wioleta Pietrzak, Katarzyna Klimek, Anna Grzywa-Celińska, Rafał Celiński, Marek Gogacz

**Affiliations:** 1Department of Pharmaceutical Botany, Medical University of Lublin, 20-093 Lublin, Poland; wioleta.pietrzak@umlub.pl; 2Department of Biochemistry and Biotechnology, Medical University of Lublin, 20-093 Lublin, Poland; katarzyna.klimek@umlub.pl; 3Chair and Department of Pneumonology, Oncology and Allergology, Medical University of Lublin, 20-090 Lublin, Poland; annagrzywacelinska@umlub.pl; 4Department of Cardiology, Independent Public Provincial Specialist Hospital in Chełm, 22-100 Chełm, Poland; rcelinski@op.pl; 5II Chair and Department of Gynecology, Medical University of Lublin, 20-090 Lublin, Poland; gogacz@yahoo.com

**Keywords:** *Alchemilla*, Rosaceae, LC-MS, antioxidant, cytotoxicity

## Abstract

Liquid chromatography electrospray ionization tandem mass spectrometric (LC-ESI-MS/MS) qualitative and quantitative analysis of different extracts from the aerial parts and roots of *Alchemilla acutiloba* led to the identification of phenolic acids and flavonoids. To the best of our knowledge, isorhamnetin 3-glucoside, kaempferol 3-rutinoside, narcissoside, naringenin 7-glucoside, 3-*O*-methylquercetin, naringenin, eriodictyol, rhamnetin, and isorhamnetin were described for the first time in *Alchemilla* genus. In addition, the antioxidant, anti-inflammatory and cytotoxic activity of all extracts were evaluated. The results clearly showed that among analyzed extracts, the butanol extract of the aerial parts exhibited the highest biological activity comparable with the positive controls used.

## 1. Introduction

The genus *Alchemilla* includes a large number of types in the Polish flora that are not easy to identify. *Alchemilla* species are apparently very similar to each other, but apart from the morphological diversity, they differ remarkably in terms of ecology, phytosociology and geography.

*Alchemilla* herb has long been used in European and Asian folk medicine in diseases caused by poor metabolism as well as for the treatment of eczema, wounds, ulcers,, and gynaecological problems [1,2,3]. Due to the relatively high tannin content, the *Alchemilla* herb has an effect characteristic of this group of compounds. After oral administration of the extract, diarrhea is reduced or inhibited [4]. The astringent properties of tannins impede the penetration of fluid into the intestinal lumen, as well as stop bleeding from damaged capillaries, reduce inflammation of the mucous membranes of the gastrointestinal tract, and inhibit the growth of pathogenic bacteria strains [4]. *Alchemilla* herb extract also acts on the skin and connective tissue, especially in damaged areas, such as scars after wounds and burns. Sometime after application, partial regeneration of the capillaries is observed, the spots and traits on the skin slowly disappear, and the normal immunity and elasticity of the epidermis are restored in these places [3,4].

*Alchemilla* herb extracts have been used in chronic gastric and intestinal disorders with symptoms of abdominal pain, loss of appetite, vomiting, and diarrhea. The herb is used externally for skin damage and inflammation (compresses), vulvar itching, vaginal discharge and vaginitis (irrigations), and mild diseases of the mouth and throat. In addition, an infusion or decoction of the herb of *Alchemilla* is used in the form of compresses and washes in conjunctivitis [4,5].

Earlier studies of the chemical composition of *Alchemilla* usually concerned the broadly understood species of *Alchemilla vulgaris*. Previous findings showed that *Alchemilla* species comprise mostly phenolic compounds such as tannins, phenolic acids, flavonoids [6,7,8,9], as well as essential oils [10] and fatty acids [11,12]. 

Extraction is one of the most important steps in receiving active compounds from medicinal plants. The biological quality of plant-derived extracts is based on the contents of the active compounds, and various solvents can extract various amounts of active compounds due to distinct affinities for specific groups of active compounds [13].

To the best of our knowledge, there are no such studies of *Alchemilla acutiloba*. Due to the importance of *Alchemilla* species in traditional medicine, the purpose of the present research was to evaluate the biological properties of different extracts of the aerial parts and roots of *A. acutiloba* and conduct their qualitative and quantitative analysis using LC-ESI-MS/MS. 

## 2. Results and Discussion

### 2.1. Phytochemical Analysis

Total phenolic content (TPC) for *A. acutiloba* extracts and subfractions was estimated using Folin–Ciocalteu reagent, and the results were expressed as gallic acid equivalents (GAE) per g of dry extract (DE) (Table 1). The results showed that the aerial parts (AH) have the highest phenolic content (279.82 ± 1.52 mg GAE/g DE) than the roots (AR) (192.54 ± 0.49 mg GAE/g DE). The results obtained in our study were better compared to data presented in previous research of *Alchemilla* genus. For instance, Vitkova et al. [6] demonstrated that TPC for ethanolic extracts of the aerial parts of thirteen *Alchemilla* species belonging to different sections varied from 33.75 to 82.71 mg GAE/g of dry extract. Neagu and co-authors [14] also found lower amounts of polyphenols in aqueous and 70% ethanolic extracts of the herb of *A. vulgaris* (94.66 and 112.33 μg GAE/mL of extract). On the other hand, Boroja et al. [15] found 558.19 mg GAE/g of dry extract total phenolic compounds in methanolic extract of the aerial parts of *A. vulgaris* and 442.32 mg GAE/g of dry extract in extract of the roots, and these values are almost twofold greater than our results.

The content of polyphenols was also examined in fractions obtained after fractionating of crude extracts between solvents of different polarity. The results showed that the TPC level was in the range of 93.15 ± 0.95 (ethyl acetate fraction from the roots) to 215.61 ± 2.10 mg GAE/g DE (butanol fraction from the aerial parts). Karatoprak et al. [16], who also studied amounts of phenolics in different fractions of *A. mollis*, obtained lower values with the highest TPC level for butanol fraction (63.50 mg GAE/g extract). 

Thus, the results obtained in our study indicated that the aerial parts of *A. acutioloba* are a rich source of polyphenols. 

The total flavonoid content (TFC) was also studied using the previously described colorimetric method [17]. The data were expressed as quercetin equivalents (QE) per g of dry extracts (DE). The results presented in Table 1 showed that the higher content of TFC was noticed for ethyl acetate fraction and crude extracts of the aerial parts (189.25 ± 0.95 and 113.79 ± 1.09 mg QE/g DE, respectively). The lowest content was noted for diethyl ether fractions of the aerial parts and roots (1.57 ± 0.02 and 1.92 ± 0.01 mg QE/g DE, respectively). The results obtained in our study were higher than those described by Karatoprak and co-authors [16]. In their research of the aerial parts of *A. mollis*, quantitative estimation revealed that methanol extract possessed 50.63 ± 0.59 mg CA/g DE flavonoid content followed by water extract—47.34 ± 0.39 mg CA/g DE, and 70% methanol—34.63 ± 0.59 mg CA/g DE. Flavonoids were not found in the hexane and ethyl acetate fractions. Also, a lower content of flavonoids was found in the aerial parts and roots of *A. vulgaris* (13.30 ± 1.69 and 19.80 ± 0.35 mg RUEs/g, respectively) [15]. 

The total phenolic acids content (TPAC) in *A. acutiloba* extracts and fractions were presented in Table 1. The amounts of TPAC found were relatively low. A higher content was noted for butanol fraction and crude extract of the aerial parts (72.19 ± 0.39 and 39.15 ± 0.18 mg CAE/g DE, respectively). TPAC was not found in the diethyl ether fraction of the aerial parts of *A. acutiloba*. Boroja et al. [15] found that methanolic extracts of the aerial parts and roots of *A. vulgaris* contain 33.43 and 57.36 mg CAE/g DE, respectively.

### 2.2. Qualitative and Quantitative Analysis

The next stage of our study was qualitative and quantitative analysis of flavonoids and phenolic acids in the crude extracts and fractions from the aerial parts and roots of *A. acutiloba*. The optimized LC-ESI-MS/MS procedure allowed us to identify the largest amount of phenolic compounds in the butanol extract from the aerial parts (20), and 18 from the roots. The representative chromatograms of flavonoid glycosides and phenolic acids in the ethyl acetate extract of the aerial parts of *A. acutiloba* (AH-O) are shown in Figure 1 and Figure 2, respectively. 

The concentrations of all individual compounds that were quantified by comparison of peak areas with the calibration curves obtained for the corresponding standards are shown in Table 2. The protocatechuic acid, rosmarinic acid, quercitrin, isoquercitrin, kaempferol 3-rutinoside and narcissoside were detected in quantifiable amounts in all studied extracts. The *p*-coumaric, caffeic, vanillic and 4-hydroxybenzoic acids content of the butanol extract of the aerial parts (AH-B) (187.6 ± 0.2, 78.9 ± 0.2, 76.8 ± 0.4 and 70.4 ± 0.2 μg/g DE, respectively) were found to be much higher than those of the other extracts. Caffeic and gentisic acids were previously found in the aerial parts of *A. mollis* [16]. The content of caffeic acid in 70% methanolic extract of *A. mollis* (40 μg/g DE) was higher than in our study for 60% methanolic extract (23.50 μg/g DE). On the other hand, Nikolova et al. [18] found that caffeic, protocatechuic, gentisic, and salicylic acids occurred in the greatest quantity among identified free phenolic acids in the extract of *A. jumrukczalica* and *A. vulgaris*, but amounts of these acids were lower than those received in our study. Ferulic acid was detected in quantifiable amount only in AH-B extract (15.9 ± 0.1 μg/g DE), and gallic acid only in the crude extracts from the aerial parts and roots (5.89 ± 0.05 and 2.45 ± 0.1 μg/g DE). The amount of identified flavonoid aglycones was relatively small. The highest content of these compounds was also observed in AH-B extract (26.2 μg/g DE). Among the obtained extracts, it was found that the ethyl acetate (AH-O) and the methanol (AH) extract of the aerial parts of *A. acutiloba* have the highest total flavonoid glycosides content (1291.7 and 1033.8 μg/g DE, respectively). Narcissoside, kaempferol 3-rutinoside and quercitrin were the most abundant glycosides in all extracts studied. High amounts of rutin was observed in the crude extract and the ethyl acetate fraction of the aerial parts (235.00 ± 3.54 and 332.50 ± 1.18 μg/g DE, respectively). Karatoprak et al. [16] also found a great amounts of rutin in different extracts of the aerial parts of *A. mollis*. In particular, the butanol fraction and 70% methanolic extract were rich in rutin (840 and 720 μg/g DE, respectively).

For the best of our knowledge, isorhamnetin 3-glucoside, kaempferol 3-rutinoside, narcissoside, naringenin-7-glucoside, 3-*O*-methylquercetin, naringenin, eriodictyol, rhamnetin, and isorhamnetin were described for the first time in *Alchemilla* genus. 

### 2.3. Antioxidant Activities

The antioxidant activities of various extracts of the aerial parts and roots of *A. acutiloba* were determined employing the 2,2-diphenyl-1-picrylhydrazyl (DPPH^•^) and 2,2-azino-bis-(3-ethyl-benzthia-6-sulfonic acid) (ABTS^•+^) radical scavenging assays. It was found that the radical scavenging activity depended on concentration. At a dose of 50.0 µg/mL, the DPPH^•^ scavenging abilities were the highest for the ethyl acetate (94.85 ± 0.30%) and the butanol (87.31 ± 0.17%) fraction of the aerial parts, and in the ABTS^•+^ assay for the AH-B (80.56 ± 0.17%). Our result can be compared to the radical scavenging activity of *A. jumrukczalica* and *A. vulgaris* complex (the commercial herbal mixture of *Alchemilla* species) analyzed by Nikolova et al. [18]. The extracts showed significant antiradical activity, similar to those obtained in our study, with IC_50_ values of 12.09 and 19.62 µg/mL for *A. jumrukczalica* and *A. vulgaris* complex, respectively. Approximately ten times higher IC_50_ values in the DPPH^•^ test for various *A. mollis* extracts were obtained by Karatoprak et al. [16]. The most active were 70% methanolic and water extracts with IC_50_ values 0.21 and 0.24 mg/mL, respectively. Boroja et al. [15] also studied antioxidant capacity with DPPH test of the methanolic extracts from the aerial parts and roots of *A. vulgaris*. They found that these extracts possessed a significant scavenging effect with IC_50_ value 5.96 and 11.86 µg/mL, respectively. Quite similar results were also obtained in the ABTS^•+^ assay (IC_50_ = 14.80 and 32.49 µg/mL, respectively). Among the analyzed extracts, the butanol fraction of the aerial parts of *A. acutiloba* were also the most active ones interfering with the formation of iron and ferrozine complexes (IC_50_ = 11.43 ± 0.18 µg/mL of DE), which suggest their high chelating capacity and ability to capture iron ions before ferrozine. The chelating activity of AH-B fraction was similar to the activity of the standard used (Na_2_EDTA*2H_2_O, IC_50_ = 9.45 ± 0.03 µg/mL). The IC_50_ values of antioxidant capacities are presented in Table 3.

### 2.4. Anti-Inflammatory Activity

In our study, the COX-1 and COX-2 enzymes’ inhibitory activities of extracts of *A. acutiloba* were investigated as a mechanism of their anti-inflammatory action. In the present study, for what we believe is the first time, the anti-inflammatory activity of *A. acutiloba* was investigated. The extracts were studied at two concentrations: 50 and 100 µg/mL. Indomethacin at a concentration 5 µM was used as a positive control. The results of the inhibition of both cyclooxygenases were recorded as the percentage inhibition of prostaglandin biosynthesis and are presented in Table 4. According to Eldeen et al. [19], a minimum inhibition of 50% is needed for plant extracts to be considered active.

Our results revealed that at a concentration of 50 μg/mL, AH-B, AR-B and AH-O extracts were capable of inhibiting the activity of COX-1 enzyme by 76.82%, 63.17% and 52.65%, respectively, whereas the inhibition of COX-2 was higher (79.75%, 72.64%, and 60.23%, respectively). At a concentration of 100 μg/mL, the most active against COX-1 was also AH-B (83.14%) followed by AR-B (78.29%), and AH-O (74.50%), while the weakest were diethyl ether fractions from the aerial parts and roots—AH-E (10.43%) and AR-E (28.71%). Except for AH-E and AR-E, all extracts in 100 μg/mL concentration showed good activity against COX-2 (54.25–95.10%). Therefore, our results can be of importance for further research of *A. acutiloba* as a potential anti-inflammatory treatment. Boroja and co-authors [15] evaluated the anti-inflammatory activity of the aerial parts and roots of *A. vulgaris* using COX-1 and COX-2 assays, and the assay for determination of COX-2 gene expression. Their research revealed the preferential COX-2 inhibitory activity of methanolic extract from the aerial parts of *A. vulgaris*. Trouillas et al. [20] also examined the anti-inflammatory activity of *A. vulgaris* for the inhibition of 15-lipoxygenase activity, and their results showed that the anti-inflammatory effect of *A. vulgaris* can be related to the inhibitory activity of phenolics on arachidonic acid metabolism through the lipoxygenase pathway.

### 2.5. Evaluation of Cytotoxicity

The MTT assay indicated that tested extracts obtained from the aerial parts and roots of *A. acutiloba* possessed different cytotoxicity (Figure 3). In the case of these from the aerial parts, it was demonstrated that the butanol fraction (AH-B) was not cytotoxic towards GMK cells, as the CC_50_ value was equal to approx. 1000 μg/mL. Other extracts from the aerial parts was more cytotoxic towards GMK cells. The CC_50_ values for AH, AH-O, and AH-E were approx. 360 μg/mL, 270 μg/mL, and 274 μg/mL, respectively. Interestingly, all fractions from roots (AR-B, AR-O, and AR-E) were less cytotoxic than the initial extract (AR). Thus, CC_50_ values for AR-B, AR-O, and AR-E were approx. 1000 μg/mL, 1000 μg/mL, and 235 μg/mL, while CC_50_ value for AR was equal to 104 μg/mL. Based on the obtained results, it was proved that the butanol fraction from the aerial parts (AH-B) as well as butanol and ethyl acetate fractions from the roots of *A. acutiloba* (AR-B, AR-O) were the most promising extracts.

## 3. Materials and Methods

### 3.1. Plant Material

The aerial parts and roots of *Alchemilla acutiloba* Opiz were collected near Karpacz in Poland (coordinates N 50°78′09′′; E 15°73′09′′). A voucher specimen (voucher no. AA-0818) was deposited in the Department of Pharmaceutical Botany, Faculty of Pharmacy, Medical University of Lublin. The plant species was identified by Prof. Tadeusz Krzaczek.

### 3.2. Chemicals and Reagents

Ascorbic acid, 2,2-diphenyl-1-picrylhydrazyl radical (DPPH^•^), 2,2′-azinobis-(3-ethylbenzothiazoline-6-sulfonic acid) (ABTS^•+^), disodium dihydrate (Na_2_EDTA*2H_2_O), (±)-6-hydroxy-2,5,7,8-tetramethylchromane-2-carboxylic acid (Trolox), Indomethacin were obtained from Sigma-Aldrich (Steinheim, Germany). Phosphate-buffered saline (PBS) was purchased from Gibco (Carlsbad, CA, USA). Reference compounds of phenolic acids and flavonoids were purchased from ChromaDex (Irvine, CA, USA). Acetonitrile, water and formic acid for LC analysis were from Merck (Darmstadt, Germany). All others chemicals were of analytical grade and were obtained from Polish Chemical Reagent Company (POCH, Gliwice, Poland). 

### 3.3. Extraction Procedure 

Different solvent systems (60% methanol, diethyl ether, ethyl acetate and *n*-butanol) were used to prepare the extracts and subfractions of the *A. acutiloba* aerial parts and roots. The air-dried, ground aerial parts and roots (20.0 g) were separately extracted with 60% methanol (3 × 200 mL) in an ultrasonic bath (InterSonic IS-4, Olsztyn, Poland) at a controlled temperature (40 ± 2 °C) for 45 min. Extractants were evaporated under reduced pressure to dryness under vacuum at a controlled temperature (40 ± 2 °C), and then subjected to lyophilization using a vacuum concentrator until constant weights were obtained. The obtained yields were as follows: from the aerial parts—AH—4.6 g; from the roots—AR—3.0 g. The obtained samples were re-dissolved in water and successively partitioned between diethyl ether (AH-E and AR-E), ethyl acetate (AH-O and AR-O) and *n*-butanol (AH-B and AR-B). All subfractions were concentrated to dryness under vacuum at controlled temperature. The obtained yields: AH-E—1.4 g; AR-E—0.4 g; AH-O—0.8 g; AR-O—0.6 g; AH-B—1.9 g; AR-B—1.7 g.

### 3.4. Total Flavonoid, Phenolic and Phenolic Acids Content

Total flavonoid (TFC) and total phenolic content (TPC) were established using the colorimetric assays as described previously [17]. The absorbance was measured at 430 and 680 nm, respectively, using Pro 200F Elisa Reader (Tecan Group Ltd., Männedorf, Switzerland). The results for TPC were expressed as mg of gallic acid equivalent (GAE) per 1 g of dry extract (DE), and for TFC as mg of quercetin equivalent (QE) per 1 g of DE. Total phenolic acids (TPAC) content was assessed using Arnov’s reagent as described in Polish Pharmacopoeia IX (an official translation of PhEur 7.0) [21]. The results were expressed as mg of caffeic acid equivalent (CAE) per 1 g of DE. 

### 3.5. LC-ESI-MS/MS Analysis 

An Agilent 1200 Series HPLC system (Agilent Technologies, Santa Clara, CA, USA) coupled to a 3200 QTRAP mass spectrometer (AB Sciex, Redwood City, CA, USA) was used for the analysis of phenolic acids and flavonoids in *A. acutiloba* various extracts. The separation of compounds was performed on a Zorbax SB-C18 analytical column (2.1 × 100 mm, 1.8 µm, Agilent Technologies, Palo Alto, CA, USA) at 25 °C. Elution was conducted using solvent A (0.1% HCOOH in water) and solvent B (0.1% HCOOH in acetonitrile). The following gradient elution program was used: 0–2 min—20% B, 3–4 min—25% B, 5–6 min—35% B, 5–6 min—35% B, 8–12 min—65% B, 14–16 min—80% B, 20–28 min—20% B. The flow rate was 300 µL/min. The mass spectra of analyzed compounds were acquired in the negative ESI mode, and the optimum values of the source parameters were as follows: capillary temperature 450 °C, curtain gas 30 psi, nebulizer gas 50 psi, source voltage −4500 V for phenolic acids and flavonoid glycosides, and capillary temperature 550 °C, curtain gas 20 psi, nebulizer gas 30 psi, and source voltage −4500 V for analysis of flavonoid aglycones. The other details of LC-ESI-MS/MS analysis were described in our previous research [17].

### 3.6. Antioxidant Activity

To establish the antioxidant potential of the extracts of *A. acutiloba,* three methods were applied. 2.2-diphenyl-1-picryl-hydrazyl (DPPH^•^) free radical scavenging activity, 2,2′-azinobis[3-ethylbenzthiazoline]-6-sulfonic acid (ABTS^•+^) decolorization assay, and metal chelating activity were determined using methods described previously [17].

All assays were performed using 96-well microplates (Nunclon, Nunc, Roskilde, Denmark) and were measured in an Infinite Pro 200F Elisa Reader (Tecan Group Ltd., Männedorf, Switzerland). Results were expressed as the IC_50_ values of *A. acutiloba* extracts based on concentration–inhibition curves. L-ascorbic acid, Trolox and Na_2_EDTA*2H_2_O were used as a positive control.

### 3.7. Cyclooxygenase-1 (COX-1) and Cyclooxygenase-2 (COX-2) Inhibitory Activity

The extracts of *A. acutiloba* were examined for cyclooxygenase-1 (COX-1) and cyclooxygenase-2 (COX-2) inhibitory activity using a COX (ovine/human) Inhibitor Screening Assay Kit (No. 560131, Cayman Chemical Company, Ann Arbor, MI, USA) according to the protocol of the manufacturer. The extracts were tested different concentrations. Indomethacin was used as a positive control.

### 3.8. Evaluation of Cytotoxicity

The cytotoxicity of tested extracts and fractions was assessed using representative normal cells, namely green monkey kidney cells (GMK) [22]. This cell line was purchased from BIOMED Serum and Vaccine Production Plant (Lublin, Poland). The GMK cells were cultured as described previously in details [22]. For cytotoxicity determination, GMK cells were seeded in 96-well plates at concentration of 2 × 10^4^/well and maintained for 24 h at 37 °C. Next day, the serial dilutions of tested extracts in culture medium were prepared (1000 μg–1.95 μg/mL), and then 100 μL of these solutions was added to the cells. Cells cultured without tested extracts were served as control (0 μg/mL). After 24-h incubation, the cell viability was assessed using MTT assay [22]. The results were presented as mean values ± standard deviation (SD) of three independent experiments. Statistical analysis was performed using unpaired Student’s t-test and differences were considered significant when *p* < 0.05 (GraphPad Prism 5, version 5.04 Software, GraphPad Software, San Diego, CA, USA). The values of half-maximum cytotoxic concentration (CC_50_) were determined using 4-parameter nonlinear regression analyses, GraphPad Prism 5, version 5.04. The CC_50_ denotes a concentration of extracts required for reduction of cell viability to 50%.

### 3.9. Statistical Analysis

The results were expressed as mean values ± standard deviation (SD) of three independent experiments. The data from cell culture experiments were subjected to statistical analysis using unpaired Student’s t-test and differences were considered significant when *p* < 0.05 (GraphPad Prism 5, version 5.04).

## 4. Conclusions

Medicinal plants are a great source of novel pharmaceutical products. Modern phytotherapy recommends using appropriate extraction procedures and standardization of extracts containing purified and concentrated active compounds [23,24]. To account for this fact, in our research we prepared and studied dry extracts and fractions of *A. acutiloba* leaves and roots using fractionated extraction and solvents of various polarity. The crude methanol–water (6:4, *v*/*v*) extracts, prepared with the solvent indicated previously as the most effective extractant of polyphenols, were used as the starting extracts for fractionation. The solvents were selected experimentally based on previous research on other plant materials [24,25,26]. The extraction conditions enabling the best recovery of phenolic acids and flavonoid compounds from the raw material and biological activity were selected. The fractionation procedure allowed for the enrichment of the extracts in selected analytes, e.g., phenolic acids such as caffeic acid, 4-hydroxybenzoic acid, vanillic acid, *p*-coumaric acid, salicylic acid in AH-B (butanol fraction from the aerial parts), and flavonoid glycosides such as quercitrin, isoquercitrin, kaempferol-3-rutinoside, rutin and narcissoside in AH-O (ethyl acetate fraction from the aerial parts).

In our in vitro research, we characterized polyphenol composition and evaluated biological properties of different extracts from leaves and roots of *A. acutiloba*. Thus, we identified the main polyphenols as well as determined the antioxidant, anti-inflammatory, and cytotoxic properties of these extracts.

The phytochemical investigation of the aerial parts and roots of *A. acutiloba* led to the qualitative and quantitative analysis of flavonoids and phenolic acids. Among these compounds, isorhamnetin 3-glucoside, kaempferol 3-rutinoside, narcissoside, naringenin-7-glucoside, 3-*O*-methylquercetin, naringenin, eriodictyol, rhamnetin, and isorhamnetin were described for the first time in the investigated species.

Moreover, our findings demonstrated that the butanol extract of the aerial parts of *A. acutiloba* as well as the butanol and ethyl acetate extracts of the roots exhibited strong antioxidant and anti-inflammatory activities, and were not cytotoxic towards GMK cells.

Taking into account the results of present as well as previously published studies of *Alchemilla* genus, it is reasonable to conclude that *A. acutiloba* is an abundant source of secondary metabolites that benefit health. It seems to be clear that comprehensive and well-designed future research on phenolic compounds from *A. acutiloba* will be of significant importance in pharmacy and medicine.

## Figures and Tables

**Figure 1 molecules-27-00621-f001:**
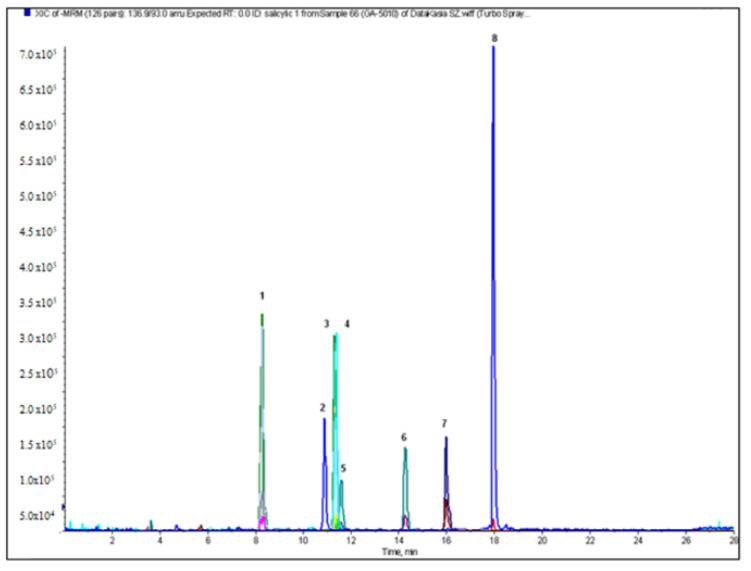
The chromatogram in MRM mode of phenolic acids in the ethyl acetate extract of the aerial parts of *A. acutiloba* (AH-O): 1-protocatechuic acid; 2-4-hydroxybenzoic acid; 3-gentisic acid; 4-caffeic acid; 5-syringic acid; 6-*p*-coumaric acid; 7-rosmarinic acid; 8-salicylic acid.

**Figure 2 molecules-27-00621-f002:**
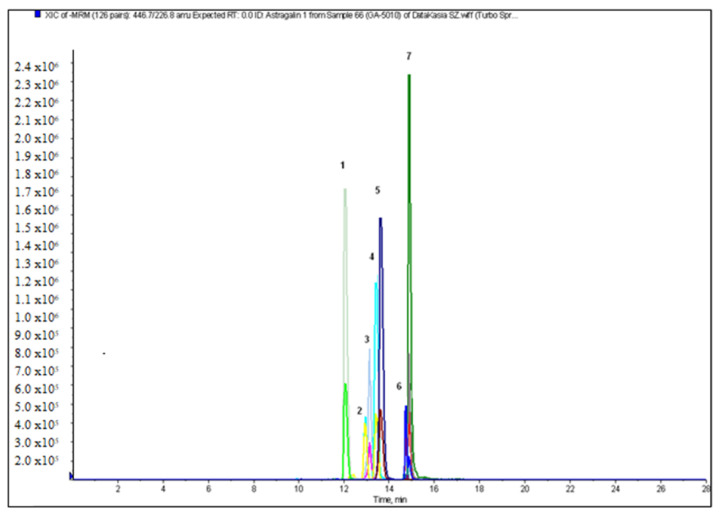
The chromatogram in MRM mode of flavonoid glycosides in the ethyl acetate extract of the aerial parts of *A. acutiloba* (AH-O): 1-rutin (quercetin-3-*O*-rutinoside); 2-isoquercetin (quercetin-3-*O*-glucoside); 3-kaempferol-3-*O*-rutinoside; 4-narcissoside (isorhamnetin-3-*O*-rutinoside); 5-astragalin (kaempferol-3-*O*-glucoside); 6-isorhamnetin-3-*O*-glucoside; 7-quercitrin (quercetin-3-*O*-rhamnoside).

**Figure 3 molecules-27-00621-f003:**
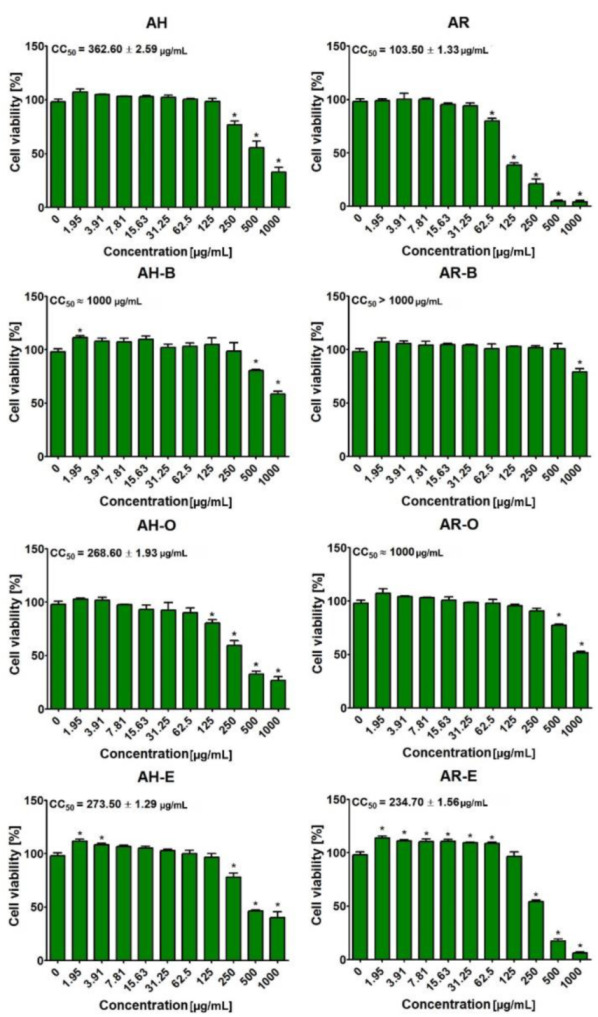
Cell viability after 24-h treatment with AH, AR, AH-B, AR-B, AH-O, AR-O, AH-E, and AR-E compounds. * Significantly different results compared to control cells (incubated without tested compounds—0 μg/mL); unpaired t-test, *p* < 0.05.

**Table 1 molecules-27-00621-t001:** The total content of phenolic (TPC), flavonoid (TFC), and phenolic acids (TPAC) in the aerial parts and roots of *A. acutiloba*.

Sample	Total Flavonoid Content [mg QE/g DE]	Total Phenolic Content [mg GAE/g DE]	Total Phenolic Acids [mg CAE/g DE]
AH	113.79 ± 1.09	279.82 ± 1.52	39.15 ± 0.18
AH-B	57.09 ± 0.31	215.61 ± 2.10	72.19 ± 0.39
AH-O	189.25 ± 0.95	154.29 ± 1.09	11.09 ± 0.01
AH-E	1.57 ± 0.02	95.72 ± 0.89	nd
AR	19.85 ± 0.19	192.54 ± 0.49	6.52 ± 0.02
AR-B	4.63 ± 0.10	175.19 ± 1.15	9.32 ± 0.03
AR-O	23.18 ± 0.50	93.15 ± 0.95	2.15 ± 0.10
AR-E	1.92 ± 0.01	128.37 ± 1.10	5.89 ± 0.05

DE—dry extract; nd—not detected; GAE—Gallic Acid Equivalent; CAE—Caffeic Acid Equivalent; QE—Quercetin Equivalent; AH—60% methanolic extract from the aerial parts; AR—60% methanolic extract from the roots; AH-B—butanol fraction from the aerial parts, AR-B—butanol fraction of the roots, AH-E—diethyl ether fraction from the aerial parts, AR-E—diethyl ether fraction from the roots, AH-O—ethyl acetate fraction from the aerial parts, AR-O—ethyl acetate fraction from the roots. Values were presented as mean ± standard deviation (*n* = 9).

**Table 2 molecules-27-00621-t002:** Content [μg/g DE] of phenolic acids and flavonoids in the aerial parts and roots of *A. acutiloba*.

Compound	AH	AR	AH-B	AR-B	AH-O	AR-O	AH-E	AR-E
gallic acid	5.89 ± 0.05	2.45 ± 0.10	<LOQ	<LOQ	<LOQ	nd	0.00	0.00
protocatechuic acid	22.13 ± 0.18	0.66 ± 0.02	32.88 ± 0.18	5.71 ± 0.01	19.42 ± 0.12	0.80 ± 0.00	0.87 ± 0.01	2.41 ± 0.03
caffeic acid	23.50 ± 0.18	0.00	78.88 ± 0.18	9.71 ± 0.10	9.82 ± 0.02	<LOQ	0.28 ± 0.00	1.26 ± 0.01
syringic acid	4.98 ± 0.11	0.00	13.01 ± 0.02	1.81 ± 0.13	4.40 ± 1.25	0.00	0.00	<LOQ
4-hydroxybenzoic acid	12.81 ± 0.09	<LOQ	70.36 ± 0.18	13.04 ± 0.05	5.58 ± 0.28	< LOQ	1.43 ± 0.02	2.16 ± 0.04
vanilic acid	12.56 ± 0.70	0.00	76.75 ± 0.35	9.32 ± 0.05	<LOQ	0.00	0.00	0.00
gentisic acid	0.00	<LOQ	<LOQ	<LOQ	3.10 ± 0.05	0.24 ± 0.01	0.00	0.00
sinapic acid	0.00	0.00	<LOQ	<LOQ	0.00	0.00	0.00	0.00
*p*-coumaric acid	20.88 ± 0.18	<LOQ	187.63 ± 0.18	24.79 ± 0.20	4.26 ± 0.01	<LOQ	1.84 ± 0.01	4.47 ± 0.05
ferulic acid	0.00	0.00	0.00	15.93 ± 0.10	0.00	0.00	<LOQ	<LOQ
rosmarinic acid	7.56 ± 0.52	0.14 ± 0.00	0.84 ± 0.01	0.02 ± 0.00	5.87 ± 0.02	0.31 ± 0.01	0.04 ± 0.00	0.05 ± 0.00
salicylic acid	11.59 ± 0.05	<LOQ	55.50 ± 0.35	6.01 ± 0.02	7.52 ± 0.12	<LOQ	0.17 ± 0.01	0.63 ± 0.00
**Total of Phenolic Acids**	**121.90**	**3.25**	**515.85**	**86.34**	**59.97**	**1.35**	**4.63**	**10.98**
astragalin	23.75 ± 0.18	<LOQ	7.48 ± 0.11	0.33 ± 0.04	22.42 ± 0.35	<LOQ	0.03 ± 0.00	2.25 ± 0.02
quercitrin	294.38 ± 4.12	1.19 ± 0.02	395.00 ± 3.54	57.36 ± 0.10	335.83 ± 1.18	4.02 ± 0.19	10.24 ± 0.12	18.58 ± 0.02
isoquercitrin	139.38 ± 0.88	1.21 ± 0.03	22.59 ± 0.16	0.66 ± 0.02	147.75 ± 1.06	5.86 ± 0.03	1.67 ± 0.02	4.85 ± 0.02
isorhamnetin-3-glucoside	4.36 ± 0.10	<LOQ	<LOQ	<LOQ	2.31±0.01	<LOQ	<LOQ	<LOQ
kaempferol-3-rutinoside	191.88 ± 0.88	37.74 ± 0.07	7.41 ± 0.16	0.63 ± 0.10	244.17 ± 3.54	53.91 ± 0.04	8.35 ± 0.00	12.82 ± 0.16
rutin	235.00 ± 3.54	83.16 ± 0.00	5.05 ± 0.60	< LOQ	332.50 ± 1.18	97.94 ± 0.42	15.99 ± 0.08	24.97 ± 0.14
narcissoside	145.00 ± 3.54	47.71 ± 0.04	5.21 ± 0.12	0.58 ± 0.01	206.67 ± 4.71	61.76 ± 0.00	8.05 ± 0.02	13.45 ± 0.07
hyperoside	0.00	0.00	2.47 ± 0.08	0.66 ± 0.02	0.00	0.00	0.00	1.52 ± 0.01
tiliroside	0.00	0.00	<LOQ	<LOQ	0.00	0.00	0.00	0.00
naringenin-7-glucoside	<LOQ	0.00	0.00	0.00	<LOQ	0.00	0.00	0.00
**Total of Flavonoid Glycosides**	**1033.75**	**171.01**	**445.21**	**60.22**	**1291.65**	**223.49**	**44.33**	**78.44**
luteolin	0.05 ± 0.00	0.00	0.51 ± 0.00	0.02 ± 0.00	<LOQ	0.00	<LOQ	<LOQ
kaempferol	0.00	0.00	1.23 ± 0.00	<LOQ	0.00	0.00	0.00	0.04 ± 0.00
3-*O*-methylquercetin	0.00	0.00	<LOQ	0.00	0.00	<LOQ	0.00	<LOQ
naringenin	<LOQ	0.00	<LOQ	0.00	<LOQ	0.00	0.00	0.00
eriodictyol	<LOQ	0.00	0.17 ± 0.00	<LOQ	<LOQ	0.00	0.00	0.00
quercetin	0.15 ± 0.00	<LOQ	22.38 ± 0.39	1.04 ± 0.01	0.14 ± 0.00	<LOQ	0.00	0.00
isorhamnetin	<LOQ	0.00	1.89 ± 0.00	0.04 ± 0.00	<LOQ	0.00	0.00	0.00
rhamnetin	0.00	0.00	<LOQ	0.00	0.00	0.00	0.00	0.00
**Total of Flavonoid Aglycones**	**0.20**	**0.00**	**26.18**	**1.10**	**0.14**	**0.00**	**0.00**	**0.04**

AH—60% methanolic extract of the aerial parts, AR—60% methanolic extract of the roots, AH-B—butanol fraction from the aerial parts, AR-B—butanol fraction from the roots, AH-O—ethyl acetate fraction from the aerial parts, AR-O—ethyl acetate fraction from the roots, AH-E—diethyl ether fraction from the aerial parts, AR-E—diethyl ether fraction from the roots. Values are presented in means ± SD, *n* = 3; nd—not detected; LOQ—limit of quantification.

**Table 3 molecules-27-00621-t003:** The DPPH^•^ and ABTS^•+^ radical scavenging activity, and metal chelating (CHEL) activity of various extracts of *A. acutiloba*.

	AH	AR	AH-B	AR-B	AH-O	AR-O	AH-E	AR-E	AA	Trolox	Na_2_EDTA*2H_2_O
**DPPH**	18.69 ± 0.04	29.87 ± 0.15	8.96 ± 0.10	12.08 ± 0.18	8.83 ± 0.37	15.37 ± 0.19	41.46 ± 0.32	51.42 ± 0.18	4.90 ± 0.09	-	-
**ABTS**	6.17 ± 0.24	14.29 ± 0.06	1.42 ± 0.18	8.78 ± 0.01	6.54 ± 0.03	10.39 ± 0.15	16.28 ± 0.15	24.82 ± 0.20	-	3.07 ± 0.01	-
**CHEL**	21.60 ± 0.39	25.76 ± 0.03	11.43 ± 0.18	12.33 ± 0.33	18.89 ± 0.94	19.30 ± 0.22	25.51 ± 0.89	44.12 ± 0.24	-	-	9.45 ± 0.03

AH—60% methanolic extract of the aerial parts, AR—60% methanolic extract of the roots, AH-B—butanol fraction from the aerial parts, AR-B—butanol fraction from the roots, AH-O—ethyl acetate fraction from the aerial parts, AR-O—ethyl acetate fraction from the roots, AH-E—diethyl ether fraction from the aerial parts, AR-E—diethyl ether fraction from the roots. The results are expressed as IC_50_ in µg/mL of DE (dry extract). Ascorbic acid (AA), Trolox, and Na_2_EDTA*2H_2_O were used as the positive control. Each value is the mean ± SD (*n* = 5).

**Table 4 molecules-27-00621-t004:** Inhibition of COX-1 and COX-2 activity of *A. acutiloba* aerial parts and roots extracts.

	COX-1 Inibition [%] ±SD		COX-2 Inhibition [%] ±SD	
Extract	50 µg/mL	100 µg/mL	50 µg/mL	100 µg/mL
AH	50.17 ± 0.85	71.32 ± 2.73	43.65 ± 1.36	78.52 ± 2.18
AR	31.27 ± 0.76	47.80 ± 1.12	48.59 ± 2.19	54.25 ± 1.23
AH-B	76.82 ± 0.69	83.14 ± 1.08	79.75 ± 1.29	95.10 ± 1.81
AR-B	63.17 ± 1.28	78.29 ± 2.25	72.64 ± 1.93	90.93 ± 2.65
AH-O	52.65 ± 2.19	74.50 ± 1.29	60.23 ± 0.67	80.12 ± 1.73
AR-O	45.73 ± 1.45	64.49 ± 2.74	58.08 ± 1.79	79.15 ± 3.10
AH-E	nd	10.43 ± 0.19	nd	28.09 ± 0.13
AR-E	nd	28.71 ± 0.35	nd	41.96 ± 1.30

AH—60% methanolic extract of the aerial parts, AR—60% methanolic extract of the roots, AH-B—butanol fraction from the aerial parts, AR-B—butanol fraction from the roots, AH-O—ethyl acetate fraction from the aerial parts, AR-O—ethyl acetate fraction from the roots, AH-E—diethyl ether fraction from the aerial parts, AR-E—diethyl ether fraction from the roots. Indomethacin (5 µM)—78.65 ± 1.28% (COX-1); 91.07 ± 2.45% (COX-2)].

## Data Availability

Data available on request.

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
