# Peer review of "LC-ESI-MS/MS Identification of Biologically Active Phenolics in Different Extracts of Alchemilla acutiloba Opiz"

_molecules, 2022, doi:10.3390/molecules27030621_

Round 1

Reviewer 1 Report

The article by Szewczyk et. al. analyzed and identified  biologically active phenolics from the extractions of Alchemilla acutiloba Opiz using LC-ESI-MS/MS spectroscopy, several compounds were first time identified in this specie. Following studies showed the extracts were generally non cytotoxic at low concentrations and demonstrated good antioxidant and anti-inflammatory activities. The introduction of LC-ESI-MS/MS spectroscopy into phytochemistry would greatly enhance the ability and efficiency to identify chemicals and thus should receive certain interests from the field. Some minor comments are provided below:

1) in all the four tables, the authors put all the annotations in the table captions, which made the table captions very bulky. It is suggested to put all the notations underneath the tables and leave the table captions alone;

2) in line 157, "....., the presence of identified in our study compounds,", the sentence is not clear, please revise;

3) in figure 3 and some parts in the main text, the volume unit "mL" was mistakenly used as "ml", please fix.

Reviewer 2 Report

The manuscript is about a fine phytochemical characterization of different extracts of Alchemilla acutiloba, a spontaneous herb collected and identified in Lublin, Poland. Based on previous literature, several species belong to the Alchemilla genus and many studies are present about their different biological acitivities. In the present work, the authors speculate on the quali-quantitative evaluation of phenolics of a single species, Alchemilla acutiloba. Moreover, antioxidant activity of the obtained extracts and their citotoxicity have been evaluated. The manuscript is well written and experimental procedures well described. Anyway, I recommend to revise the manuscript, based on the following points:

  • Different solvents have been used to obtain crude extracts from both aerial and root plant materials. The authors should better explain the scientific basis of their choices (60% MeOH, Diethyl ether, ethylacetate, n-butanolo). I think it must be related to the relative affinity to the different investigated classes of compounds but it must be stressed in the manuscript.
  • Antioxidant assays and also citotoxicity tests have been performed on the different extracts. Anyway, I have major concern on this. When an analytical characterization is concerned, a solvent extraction with not-food grade solvents is justified, but when cell viability tests are performed, it is recommended the use of food-grade solvents. Citotoxicity tests are performed with the aim of evaluating if a crude extract could be used as a phytochemical agent for the treatment of a specific disease or alteration. In this case, only crude or purified extracts obtained by using food-grade solvents should be used, because their potential employment in phytotherapy could not be otherwise recommended. Among the solvents employed by the authors, they must stress which of them had food-grade purity. The others cannot be considered for cell viability assays.
  • Conclusion section is too small. The authors should better include more discussion, also providing more details on the potential use of this kind of herb.
  • minor english grammar check is needed
